METHODS AND RESOURCES

# Comparative gene annotation and orthology assignments across 301 species of Drosophilidae

**Pankaj Dhakad**[1,2]*, **Bernard Y. Kim**[3], **Dmitri A. Petrov**[4,5], **Darren J. Obbard**[1]

**1** Institute of Ecology and Evolution, University of Edinburgh, Edinburgh, United Kingdom, **2** Evolution and Ecology, University of California, Davis, California, United States of America, **3** Department of Ecology and Evolutionary Biology, Princeton University, Princeton, New Jersey, United States of America, **4** Department of Biology, Stanford University, Stanford, California, United States of America, **5** CZ Biohub, Investigator, San Francisco, California, United States of America

* Pankaj.Dhakad@ed.ac.uk

## Abstract

High-quality genome annotations are essential if we are to address central questions in comparative genomics, such as the origin of new genes, the drivers of genome size variation, and the evolutionary forces shaping gene content and structure. Here, we present protein-coding gene annotations for 301 species of the family Drosophilidae, generated using the Comparative Annotation Toolkit (CAT) and BRAKER3, and incorporating available RNA-seq and protein evidence. We take a comparative phylogenetic approach to annotation, with the aim of improving consistency and accuracy, and to generate a robust set of gene annotations and orthology assignments. We analyze our annotations using a phylogenetic mixed-model approach and find that gene number and CDS length exhibit moderate phylogenetic heritability (40% and 9.7%, respectively). For comparison, we also present analyses using a subset of the 215 highest quality genomes, although the findings were not markedly different. Our work suggests that while evolutionary history contributes to variation in these traits, species-specific factors—including assembly error—play a substantial role in shaping observed differences. To illustrate the utility of our annotations for comparative analyses, we investigate codon usage bias and amino acid composition across Drosophilidae. We find that codon usage is correlated with overall GC content and evolves slowly, but that it is also strongly shaped by selection—such that, in general, species with the strongest selection on synonymous codon usage show the lowest GC bias in third codon positions. This comparative annotation dataset forms part of an ongoing collaborative project to sequence and annotate all species of Drosophilidae, with data and annotations being made rapidly and freely available on an ongoing basis. We hope that this effort will serve as a foundation for studies in evolutionary and functional genomics and comparative biology across Drosophilidae.

**Data availability statement:** All code used for data processing, generating figures, and statistical analyses is publicly available from Zenodo (https://doi.org/10.5281/zenodo.18444670). The *Drosophila* genome annotations and orthology assignments generated in this study are also publicly available via Zenodo (https://doi.org/10.5281/zenodo.15016917).

**Funding:** This work was supported by a Darwin Trust PhD studentship to PD (https://biology.ed.ac.uk/darwintrust), a Biotechnology and Biological Sciences Research Council grant BB/T007516/1 (https://www.ukri.org/councils/bbsrc/) to DJO, and a National Institutes of Health, National Institute of General Medical Sciences grant R35 GM118165 (https://www.nigms.nih.gov/) to DAP. DAP is also supported as a Chan Zuckerberg Biohub Investigator. PD received a salary in the form of a PhD studentship from the Darwin Trust. DJO and DAP received research funding support from their respective grants listed above. The funders had no role in study design, data collection and analysis, decision to publish, or preparation of the manuscript.

**Competing interests:** The authors have declared that no competing interests exist.

**Abbreviations:** BUSCO, Benchmarking Universal Single-Copy Orthologs; CAT, Comparative Annotation Toolkit; CDS, coding sequence; GC3, GC content at third codon positions; GLMM, Generalized Linear Mixed Model; HOGs, Hierarchical Orthologous Groups; HPD, highest posterior density; MRCA, most recent common ancestor; Mya, Million years ago; N/C, nitrogen-to-carbon; OGs, orthogroups; PCA, principal component analysis; PGLMM, Phylogenetic Generalized Linear Mixed Model; TEs, transposable elements.

## Introduction

Fundamental questions in comparative genomics include the origin of new genes, the causes of genome size variation, and the factors that shape rates of genome evolution. However, addressing these questions requires not just the genome sequence, but also accurate identification and characterization of genomic features such as coding DNA sequence. High-quality genome annotations are thus essential for the identification of loci that underpin phenomena such as adaptation and speciation [1]. However, the annotation of genomes remains challenging, due to complex gene structures, long non-coding regions, and species-specific features [1–3]. Automated annotation methods, while efficient, often produce artifacts that can mislead interpretations of gene function and evolutionary relationships [4–6]. For example, an early annotation of the *Daphnia pulex* genome may have over-estimated the number of genes and over-predicted paralogs of genes involved in environmental responsiveness, potentially leading to initial misinterpretations of the basis of its adaptive capabilities [7,8].

Over the past decade, advances in long-read sequencing technologies and scaffolding methods, together with the declining cost of sequencing, have led to a dramatic increase in the number and quality of genome assemblies across the tree of life [9,10]. This surge is exemplified by large-scale initiatives such as the Darwin Tree of Life project [11], which aims to sequence around 70,000 species in the UK and Ireland, the Vertebrate Genome Project [12], which has the goal of sequencing all extant vertebrate species, and the African BioGenome Project [13], which seeks to sequence more than 105 thousand species in Africa. However, despite advances in sequencing and assembly, genome annotation remains a major bottleneck [14]. Most genomes from large-scale sequencing projects are annotated independently using automated pipelines such as BRAKER, Augustus, and MAKER [15–17]. While these annotations can incorporate evidence from reference protein databases (e.g., OrthoDB), they typically do not exploit whole-genome alignments across multiple closely related species or conserved gene order (synteny) to improve annotation accuracy and consistency across genomes [18,19]. As a result, different pipelines can identify slightly different sets of genes due to variation in prediction algorithms, parameter settings, and assumptions about gene structure [20]. This inconsistency means that genes missed by one pipeline might be detected by another, while certain gene models may only partially align between annotations [14]. Consequently, such independent annotations make it difficult to achieve a standardized gene set for comparative analyses [7,8,20]. Comparative annotation, on the other hand, aims to address these issues by using alignment with well-annotated reference genomes to help guide predictions, reducing discrepancies and aligning gene models more consistently across closely related species [21,22]. This approach can not only increase the accuracy of gene predictions, but can also ensure a more robust, comparable gene set for evolutionary studies. The Comparative Annotation Toolkit (CAT) is one such method, designed to annotate genomes by projecting known gene models from a reference genome onto target genomes within a phylogenetic framework [22]. CAT integrates evidence from such a "lift-over" with short and long read RNA-seq data,

Iso-seq data, and protein alignments to refine gene model predictions, weighting features that are shared among close relatives more heavily.

For over a century, *Drosophila melanogaster* and its relatives have been at the forefront of genetics, genomics, and evolutionary research, leading to influential discoveries that have shaped these fields [23]. High-quality genome assemblies and annotations have been developed for several key species, beginning with the pioneering sequencing of the *Drosophila melanogaster* genome, which served as a reference for subsequent genomic studies [24,25]. The *Drosophila* 12 Genomes Project expanded this foundation, offering comparative insights across multiple species, while initiatives such as the ModENCODE project further enriched our knowledge with detailed transcriptomic and epigenomic data [26,27]. Individual research groups have continued to sequence target species ([28–31], such as *D. miranda*, *D. guanche*, etc.), making possible resources such as DrosOMA (https://drosoma.dcsr.unil.ch/), which provides genus-wide orthology information for 36 *Drosophila* species [32]. This progress has now culminated in the ongoing community effort to achieve a comprehensive genomic study of the entire family Drosophilidae [33,34]. This effort includes the de novo sequencing of new species, scaffolding and improvement of existing genomes, and the generation of new transcriptomic data [33]. As of April 2024, around 360 different drosophilid species had been sequenced to varying levels of completeness—some fragmentary, from short-read data alone (e.g., *Drosophila setifemur* and *Drosophila ironensis*; [31]), but many to chromosome-level assemblies, using long-read data and/or scaffolding information from HiC (e.g., *Chmomyza fuscimana*; [35]).

Here, we contribute to this continuing community effort by providing a comparative coding-sequence annotation for 301 drosophilid genomes. We do this using a combination of CAT and BRAKER3, combining publicly available RNAseq data and previous RefSeq reference genomes [15,22]. We use phylogenetic linear mixed models to assess and compare the annotations, and as an example of the utility of our comprehensive annotation, we analyze codon and amino-acid usage bias across the family. To facilitate its use in both single-gene and genome-wide studies, the annotation is made freely available in the form of genome annotation files and also as aligned (and optionally masked) orthology groups, with annotations linked to gene orthology. This is available as two complementary datasets; the first is a complete set of 301 species selected on the basis of a more permissive quality filter, most suitable for single-gene studies of sequence evolution (e.g., identifying conserved regions, protein structure prediction, or site-level evolutionary inference), the second is a more stringently-filtered subset of 215 species, which may be more robust for studying micro-synteny, gene family evolution, or copy-number variation. In the future, we plan to continue updating this resource with regular new releases as new genomes become available. We hope that this will be a key resource, enabling gene-based analyses of evolution within this important model system.

## Methods

### Genome assemblies

We selected an initial candidate set of genome assemblies by supplementing those of Kim and colleagues [33] with all other publicly available drosophilid genomes available as of February 2024. This included genomes available in the RefSeq database release 222 [36], those generated by the Darwin Tree of Life project [11], and many assemblies generated by individual labs [37–40]. For our primary dataset, we shortlisted 301 genome assemblies that had a scaffold N50 greater than 50 Kbp and a Benchmarking Universal Single-Copy Orthologs (BUSCO) completeness score of over 90% [41], selecting the assembly with the highest scaffold N50 where multiple assemblies were available. To identify and mask repetitive elements, we used RepeatMasker v4.1.2 [42] and Dfam release 3.7 repeat library [43]. These soft-masked genomes were used for all subsequent analyses. To provide a secondary, higher-stringency dataset for comparison, we selected a subset of those genomes which had BUSCO completeness ≥97%, contig N50 ≥2 Mbp, and assemblies generated using long-read sequencing technologies (Oxford Nanopore or PacBio).

## RNA-seq and protein data

To annotate the genomes, we used the CAT [22], a pipeline that leverages external evidence ("hints") combining data such as RNA-seq, Iso-seq, proteins, and attempted lift-over from aligned references. For each of the 301 species, we first gathered available RNA-seq data to provide transcript evidence that can help resolve ambiguities in gene models. We identified suitable RNA-seq datasets using the ENA Portal API (https://www.ebi.ac.uk/ena/portal/api), selecting up to 10 paired-end RNA-seq datasets and prioritizing those with Poly-A selection to enrich mRNA [44]. Where available, we included data from up to 10 tissue types, including whole body, carcass, thorax, brain, testes, and ovaries, as well as different developmental stages and both sexes (selected SRA numbers for each species are in S1 Table). The chosen RNA-seq reads were then down-sampled and normalized to 100× coverage using BBNorm of BBMap v38.95 [45]. The normalized RNA-seq reads were aligned to their respective species genomes using the STAR v2.7.9a with default parameters [46].

To provide protein "hints," we extracted predicted protein sequences from Arthropoda using the OrthoDB v10 protein database (https://orthodb.org; [47]). Such protein sequences provide evolutionary conserved evidence that complements RNA-seq data, particularly for genes that may be underrepresented or absent in the RNA-seq datasets. We aligned these protein sequences to the genomes using miniprot v0.12 [48] with parameters "-ut8 --gtf genome_file", which are optimized for mapping proteins to genomic sequences. The alignment files generated by miniprot were then converted into hints files using the "aln2hints.pl" script from the GALBA toolkit [49].

## Reference species and cactus alignment

In addition to RNA-seq and protein hints, the CAT pipeline attempts a lift-over of annotations [22]. This uses genome-scale alignments in the Hierarchical Alignment Format [50], each comprising a single reference species and several target species. To define reference clades for annotation, we first generated a preliminary species tree including the 301 drosophilid species (plus seven outgroup species) using 1824 single-copy BUSCO loci [41]. Nucleotide sequences from each locus were aligned separately using MAFFT v7.520 [51] and used to infer a maximum likelihood (ML) gene tree using IQ-TREE v2.2.6 [52] under a GTR+I+G4 substitution model. These ML gene trees were then combined to infer a species tree using ASTRAL-III v5.15.5 [53], which aims to resolve gene-tree species tree incongruences under a model of incomplete lineage sorting.

We selected 37 "reference" species for lift-over annotations based on the completeness and quality of their genomes, as indicated by RefSeq annotations [36]. Using the ETE 3 python package [54], we applied a preorder tree traversal strategy to identify subclades that contained at least one reference species and included the most distantly related leaf within a predefined phylogenetic distance (measured as the expected number of substitutions per site). We varied the phylogenetic distance threshold between 0.005 and 0.35 to ensure each subclade included 3–15 species, with lower thresholds for densely populated regions of Drosophilidae tree and higher thresholds to include more distantly related species. In clades containing multiple potential RefSeq-annotated species, we selected a single reference species based on the similarity of its gene count to that of the best-annotated drosophilid model, *Drosophila melanogaster*. This criterion was used because the *D. melanogaster* annotation is likely to be the most complete and accurate within the family, having benefited from extensive manual curation and extensive transcriptomic and functional data [55]. This process resulted in 17 RefSeq-annotated species for use in the attempted lift-over. For species located on very long branches (i.e., divergence >0.35 from the closest potential reference) and for subclades lacking any reference-species annotations, we attempted a lift-over directly from *Drosophila melanogaster*.

We then used these "lift-over subclades" as guide trees to generate multiple whole-genome alignments with ProgressiveCactus [56]. This approach ensured that the alignments were computationally feasible, and that closely-related genomes aligned together. Finally, we employed the CAT [22] to annotate multiple target genomes simultaneously, using a lift-over of the selected reference annotation for each subclade.

## Running CAT

To perform the genome annotations, we first prepared the reference annotations and extrinsic "hints" for use in CAT [22]. RefSeq annotation files were converted using the "convert_ncbi_gff3" script provided by CAT, and the resulting GFF3 files were validated with the "validate_gff3" script to ensure compatibility. We then employed three modes of AUGUSTUS [16] in CAT: two based on transMap projections (AugustusTM/R) that project annotations from reference genomes onto target genomes, and one using ab-initio and comparative gene predictions (AugustusCGP) guided by extrinsic hints [57]. We used Comparative Gene Prediction (CGP) parameters trained on 12 well-annotated *Drosophila* species from the *Drosophila* 12 Genomes Project, based on exon and intron scoring (https://bioinf.uni-greifswald.de/augustus/datasets/; [57]).

## Complementation with BRAKER3

To complement the comparative annotations generated by CAT, and to reduce any potential reference bias, we additionally incorporated de novo non-comparative CDS predictions made by BRAKER3 [15]. BRAKER3 is an automated gene prediction pipeline that integrates RNA-seq data with gene prediction algorithms to generate gene models without reference to a reference annotation, improving the identification of novel genes and gene duplicates that may be absent from the reference. However, among the annotations generated by BRAKER3 we identified a number of transposable elements (TEs), which are not the focus of our study. Therefore, to simplify the annotation, we removed these using EarlGrey v4.1.0 [58], applying TEstrainer (https://github.com/jamesdgalbraith/TEstrainer) to ensure that recently duplicated non-TE genes were retained. Finally, to combine the BRAKER3 annotations with those from CAT, we compared the coding sequences (CDSs) of overlapping genes between the two annotation sets. For one-to-one overlapping genes, we selected the annotation with the longest CDS. In cases of one-to-many or many-to-one overlaps, we preferred the CDS annotations from CAT, as these reflect the additional support provided by comparison among relatives. Additionally, we retained all non-overlapping genes with a CDS length greater than 150 nucleotides.

## Annotation quality assessment

To assess the annotation quality, we compared our CAT-BRAKER annotations and annotations generated using BRAKER3 alone with RefSeq annotations. These comparisons focused on CDS concordance, quantified using overlap-based precision and recall (>90% CDS overlap) and overall Jaccard similarity compared to RefSeq CDSs. In addition to these direct annotation comparisons, we assessed completeness and gene repertoire quality using BUSCO and OMArk, which evaluate annotations against evolutionary expectations of conserved gene content [41,59]. BUSCO was run in protein mode using the Diptera OrthoDB dataset to estimate the proportion of conserved single-copy orthologs recovered in each annotation [60]. For OMArk, we first generated omamer search databases for each species using the 'LUCA. h5' orthology database (https://omabrowser.org/oma/current/). We then ran OMArk with taxon ID 7214 (Drosophilidae), assigning proteins to orthologous groups based on phylogenetically informed gene family classifications.

## Orthogroup assignment and CDS alignment

We identified CDS homology across the 301 *Drosophila* species and one outgroup species (*Musca domestica*) using OrthoFinder v2.5.5 [61]. OrthoFinder first identifies homology using an all-vs-all blast similarity search and then clusters sequences using a Markov clustering algorithm (MCL inflation parameter of 1.5). Subsequently, it can then identify "Hierarchical Orthologous Groups" (HOGs) that comprise the genes descended from a common ancestral gene at a specific taxonomic level—with HOGs defined for different clades nested within each other along the species phylogeny. We extracted HOGs at the level of Drosophilidae (i.e., sets of homologous sequences that have their most recent common ancestor (MRCA) at the base of Drosophilidae, or more recently) and retained for further analysis those HOGs that included at least two species and contained more than three sequences.

We aligned the sequences for each of the chosen HOGs using the aligners MACSE v2 [62] and MAFFT v7.520 [51], an approach intended to minimize the impact of any frameshifts and in-frame stop codon errors that might arise from sequencing, assembly, or annotation problems, while maximizing codon-aligned sequence length. MACSE v2 incorporates several steps to improve alignment quality, including a prefilter to remove long non-homologous insertions that may result from incorrect annotations, such as intron inclusions or alternative splicing. It then uses HMMCleaner to mask residues that appear misaligned and applies post-processing filters to mask isolated codons and patchy sequences, removing sequences if more than 80% of the residues are masked [63]. Finally, the alignments were trimmed at both ends until a nucleotide position represented by at least 70% of the sequences was reached, ensuring a high-quality alignment for downstream analyses of protein CDS. From the 301 annotated drosophilid genomes, we generated 35,642 HOGs and 22,355 high-quality alignments. In addition, we provide HOGs and corresponding alignments for a high-confidence annotation subset comprising 215 species, defined by more stringent assembly and annotation quality criteria (BUSCO completeness ≥97%, contig N50 ≥2 Mbp, estimated OMArk contamination ≤5%, and assemblies generated using long-read sequencing technologies). These HOGs form the basis of subsequent analyses of evolutionary relationships and functional conservation of genes across family Drosophilidae, and all masked and unmasked aligned sequences are made available at https://doi.org/10.5281/zenodo.15016917.

## Phylogenetic generalized linear mixed model analyses

We used Generalized Linear Mixed Model (GLMM) analyses of gene number and CDS length to identify clade-to-clade variation and outlier genomes across all 301 species, and separately across 215 species high-confidence set. Such variation may result from true evolutionary divergence, or from reference bias and the availability (or otherwise) of RNA-seq data, or from systematic errors in assembly or annotation quality. The relative impact of such factors is naturally addressed in a linear mixed model framework, treating species as a random effect and phylogenetic distance from reference species, genome size, assembly contiguity (contig N50), read-type (ONT/PacBio versus Illumina), and the availability of RNAseq as fixed-effect predictors. Because related species exhibit correlated traits (leading to pseudo-replication, if not accounted for; [64]), and because the phylogenetic correlation among related species (e.g., "phylogenetic inertia" or "phylogenetic heritability") may be of direct interest—reflecting clade-to-clade variation in gene content or the efficacy of selection—we employed a phylogenetic mixed model approach [65], implemented in the R package MCMCglmm [66]. This incorporates phylogenetic relationships to model the covariance among species, while evaluating the influence of fixed predictors such as phylogenetic distance from the lift-over reference and RNA-seq availability.

To do this, we first generated a revised species-tree topology of the 301 drosophilid species using 251 single-copy HOGs (those HOGs containing ≥300 species) employing IQTREE2 and ASTRAL-III, as for BUSCO genes above. To infer relative branch lengths in approximate time, we randomly selected 10,000 amino acid sites from the HOGs, and we used BEAST [67] to re-infer branch lengths on the (fixed) ASTRAL tree topology under a LG+G+I model with 7 gamma categories and an uncorrelated relaxed log-normal clock [68]. For the tree prior, we used birth-death process model [69], setting the fully-informative prior for the MRCA of the subgenera *Drosophila* and *Sophophora* to 47 Million years ago (Mya); 95% prior density 42–52 Mya; [34]), a uniform step prior between 0 and 1 on the birth-death growth rate, and the remaining priors to their default values. We ran the MCMC chain for $10^6$ generations sampling every 1,000 steps, and stationarity and mixing were assessed from visual inspection of the MCMC chain and Effective Sample Size in Tracer [70]. After discarding 10% of the sampled estimates as burn-in, we report divergence times as median node height for each of the clades in the summary tree.

To assess the factors influencing gene number and CDS length across drosophilid species, we fitted a multivariate phylogenetic mixed model using MCMCglmm [66]. Our model included gene number and mean CDS length as response variables, allowing us to analyze their (co-)variation with respect to predictors such as status as a lift-over reference, distance from lift-over reference, RNAseq availability, assembled genome size, assembly contig N50, and read-type. This

phylogenetic approach can help to disentangle the effects of biological and technical factors on genome annotation metrics, while accounting for phylogenetic relatedness among species. We inferred statistical "significance" on the basis of 95% highest posterior density (HPD) credibility intervals. The syntax for MCMCglmm models and priors is described in S1 File.

**Evolution of GC, codon, and amino acid composition across Drosophilidae**

Genome-wide GC content, codon usage, and amino-acid composition are shaped by a combination of mutational bias and natural selection [71,72]. To illustrate the utility of our CDS annotations and alignments for large-scale evolutionary sequence analyses, we examined the evolution of these coding-sequence traits across the 301 drosophilid species. We used the R package "cubar" [73] to calculate the overall GC content of CDSs ("GC"), and GC content at third codon positions (GC3). Additionally, we estimated the whole-genome GC content and the GC content of non-CDSs using "geecee" [74]. We used the "cusp" tool from EMBOSS [75] to calculate amino acid frequencies in each species, and the nitrogen-to-carbon (N/C) ratio for protein sequences was calculated as the weighted average of the N/C ratios of individual amino acids, with weights corresponding to the proportion of each amino acid in the sequence. We used a PCA analysis of amino acid frequencies to reduce the dimensionality.

We estimated the strength of selection on codon usage bias ($S$) using the approach of Dos Reis and Wernisch [76], which compares codon frequencies in highly expressed versus reference gene sets. To do this, we ranked *Drosophila melanogaster* genes according to their overall expression level (S2 Table; expression data obtained from FlyBase: https://flybase.org/) and analyzed the HOGs that contained these genes, assuming that the globally most highly-expressed genes in *Drosophila melanogaster* are also highly expressed in other species. As expected, these genes were dominated by those encoding ribosomal proteins, yolk proteins, salivary gland secretions, and elongation factors whose high expression is likely to be conserved. HOGs were then ranked in order of *Drosophila melanogaster* expression level and binned in 20 expression categories of *ca.* 600 genes in each category (S3 Table). Codon usage bias ($S$) was then estimated as the log-odds ratio of optimal to non-optimal codon frequencies for 2-fold degenerate codons, where the preferred codon was identified from the most highly expressed gene category [76,77]. Note that, to distinguish selection from mutational bias, this method assumes the reference and highly expressed gene sets have similar mutational patterns. Finally, we obtained bootstrap confidence intervals for $S$ by resampling genes within expression categories.

As above, we used a multivariate Phylogenetic Generalized Linear Mixed Model (PGLMM) implemented in MCMCglmm to assess the among-species phylogenetic (co-)variance in GC3, non-coding GC, estimated strength of selection on codon usage bias ($S$), observed frequencies of amino acids, and N/C ratio, while accounting for phylogenetic effects.

## Results and Discussion

### Gene annotation of 301 species

We selected 17 NCBI RefSeq annotations for use as potential lift-over references and combined this information with RNA-seq data from 91 species, protein hints (from mapping of OrthoDB proteins) from all species, and de novo prediction to annotate the remaining genomes using CAT and BRAKER3. On average, we identified 14,549 genes in each genome, with a mean CDS length of 1.60 Kbp. This is very similar to the gold-standard reference, *Drosophila melanogaster*, which currently has 13,904 protein-coding genes of mean CDS length 1.54 Kbp (Genome assembly release 6.53; GCF_000001215.4). However, there was substantial variation, reflecting both evolutionary variation among species and potential variation in genome assembly quality and RNAseq availability (Fig 1).

Most species (274 of 301) were annotated with between 12,500 and 16,000 protein-coding genes. There were three species that appear to possess more than 20,000 genes; *Drosophila vulcana* [28], *Drosophila miranda* [78], and *Drosophila punjabiensis* [28] (Fig 1 and S4 Table). In *D. miranda*, the elevated gene number is consistent with the recent evolutionary history of its sex chromosomes, namely a very young neo-sex chromosome system formed by fusion of

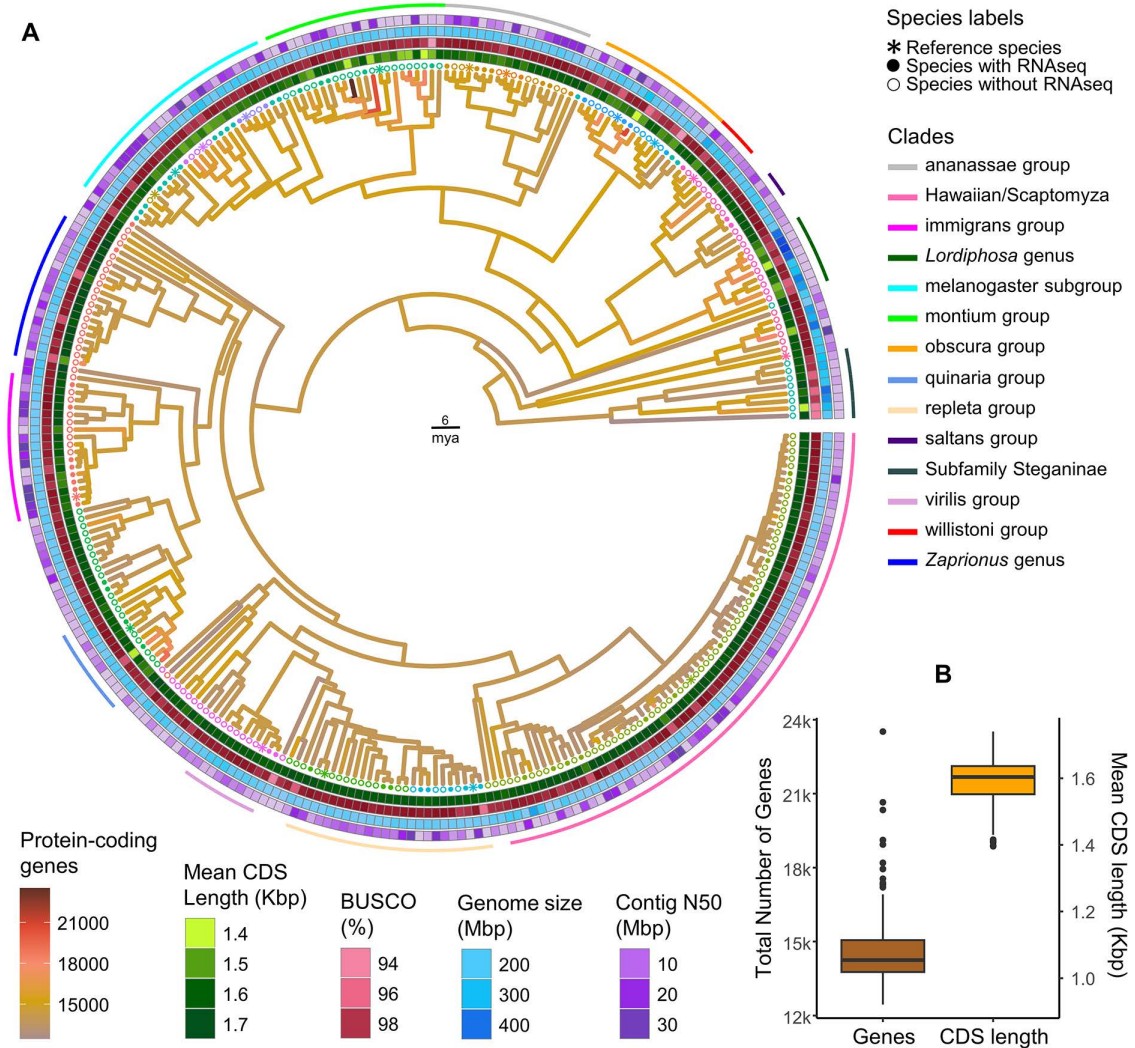

**Fig 1. Overview of 301 *Drosophila* genome annotations. (A)** Time-calibrated phylogenetic tree showing ancestral reconstruction of protein-coding gene number mapped onto branches. Tip labels are coloured by the reference species used for comparative annotation within each clade; stars denote reference species, and filled vs. open circles indicate the presence or absence of RNA-seq data, respectively. Concentric tile layers summarize key assembly and annotation metrics for each species: from the innermost to the outermost ring, mean CDS length (Kbp), BUSCO completeness (%), genome assembly size (Mbp), and contig N50 (Mbp). Major drosophilid species groups are indicated by arcs positioned outside the outermost tile layer. **(B)** The box plot represents the range in gene number and mean CDS length across family Drosophilidae. The numerical data underlying both panels are provided in S4 Table. An alternative version of the tree, with taxon labels, is provided in S2 File.

Muller element C with the ancestral Y chromosome that resulted in homologous gene copies retained on both the neo-X and neo-Y. In our annotation, we identified 3,944 genes located on the two largest neo-Y scaffolds and 2,944 genes on neo-X, consistent with extensive retention of neo-Y homologs reported previously [78]. When these neo-sex homologs are considered, the inferred gene number of *D. miranda* is closer to the genes in *D. melanogaster*, indicating that it is not anomalous in terms of gene content. In contrast, the elevated gene counts observed in *D. vulcana* and *D. punjabiensis* are more likely to reflect technical rather than biological factors (see below). The mean CDS length across species ranged between 1.40 and 1.74 Kbp. The shortest CDSs were observed in *D. miranda*, consistent with extensive fragmentation

and degeneration of amplified genes on its neo-X and neo-Y chromosomes reported previously [78]. Note that these latter two species are excluded from our subset of 215 high-confidence genomes.

Establishing whether the variation in gene number and mean CDS lengths reflects differences in annotation or assembly quality, or true evolutionary processes, is necessarily challenging in the absence of ground truth annotation for most species. We therefore assessed the quality of our CAT-based annotations using multiple complementary approaches. First, we compared CAT-BRAKER CDS annotations with available RefSeq annotations for a subset of non-*D. melanogaster* species. Using CDS overlap (>90%) as a criterion, CAT-BRAKER consistently showed higher precision relative to RefSeq than BRAKER3 alone, while recall was often slightly lower (S1 Fig). However—except perhaps in the especially well-studied case of *D. melanogaster*—RefSeq annotation should not necessarily be treated as a strict ground truth [79], and it is notable that reduced recall primarily reflects additional gene models detected by CAT and/or BRAKER that are absent from RefSeq. Consistent with this, the Jaccard similarity between CAT-BRAKER and RefSeq annotations was generally higher than between BRAKER3 and RefSeq, indicating greater overall concordance in gene content (S1 Fig).

Second, we assessed BUSCO completeness at both the genome and annotated-protein levels. For the majority of species, BUSCO completeness estimates were highly concordant between genome-based and protein-based assessments, indicating that the annotations recover most conserved genes detectable at the genome level (S2 Fig). Only *Drosophila guttifera* and *Drosophila bifasciata* showed differences greater than 5% between genome and protein BUSCO completeness (S2 Fig).

Finally, we used OMArk to further assess and compare the quality of protein-coding gene annotations. OMArk estimates the completeness, consistency, fragmentation, and contamination of gene repertoire in a species by comparison with conserved orthologous groups (HOGs). We observed high levels of HOG completeness across most species. However, *Drosophila recens* and *Drosophila miranda* showed a high number of duplications (S5 Table). In *D. miranda*, this pattern is explained by retention of homologous gene copies on the neo-X and neo-Y chromosomes (above). The source of apparent duplication in *Drosophila recens* remains unclear, but is also observed in its close relatives *D. subquinaria* and *D. suboccidentalis* (S5 Table). Additionally, OMArk identified a substantial level of contamination in the genome of *Drosophila vulcana* (5,235 of 23,519 genes identified as contaminant), a genome that is also flagged as "contaminated" in NCBI database [28]. Other genomes with apparently high levels of contamination include *Drosophila punjabiensis* (3,131 of 20,339 genes) and *Drosophila nannoptera* (2,006 of 15454 genes). Importantly, all these genomes, except *D. miranda*, showing higher duplication or contamination were excluded from the high-quality subset. Together, these assessments indicate that most annotations are highly complete at the gene level, while highlighting a small number of genomes where duplication or contamination likely inflates apparent gene counts.

## Orthology inference

We used OrthoFinder [61] to infer CDS orthology across 301 species of Drosophilidae, using *Musca domestica* as an outgroup. OrthoFinder assigned 98.6% of predicted proteins to orthogroups (OGs), with 96% further classified into HOGs. We identified a total of approximately 35 thousand HOGs across the 301 *Drosophila* species genomes (Table 1). More than 90% of genes in each species were assigned to HOGs, although some species—such as *Leucophenga varia*, *Drosophila vulcana*, *Drosophila quasianomalipes*, and *Drosophila differens*—had lower assignment rates (S4 Table and S3 Fig).

To interpret patterns of gene conservation and turnover, we classified HOGs into two broad categories: widely conserved HOGs, which include genes shared across most Drosophilidae species, and species- or clade-specific HOGs, which appear to be restricted in their distribution.

The widely conserved HOGs were further divided into "universal" HOGs, present in nearly all species (≥99%), and ancient HOGs, shared by species with MRCA of at least 50 Mya, but missing in a substantial minority of species (Table 1 and S4 Fig). Universal HOGs likely represent genes experiencing strong evolutionary constraint, encoding core biological

**Table 1. Summary statistics of HOGs in the full and high-stringency datasets.**

| Feature | Full dataset | High stringency subset |
|---|---|---|
| Number of species | 301 | 215 |
| Total number of proteins | 4,379,331 | 3,074,657 |
| Number of Orthogroups (OGs) | 29,738 | 21,946 |
| Number of proteins in OGs | 4,316,768 (98.6%) | 3,032,610 (98.6%) |
| Number of HOGs | 35,642 | 26,046 |
| Number of proteins in HOGs | 4,202,078 (96%) | 2,964,679 (96.4%) |
| Number of universal* single-copy HOG | 618 | 1,077 |
| Number of HOGs with all species present | 1,131 | 3,406 |
| Number of universal* HOGs | 6,276 | 7,732 |
| Number of HOGs containing Dmel genes | 12,151 | 12,145 |
| Number of ancient$ HOGs | 12,961 | 12,539 |

This table summarizes the composition and coverage of orthologous gene groups inferred across droso-
philid genomes using the full dataset of 301 species and a more stringently filtered subset of 215 species.
*Universal HOGs are defined as those containing genes from at least 99% of species, while $ancient HOGs
represent groups whose most recent common ancestor dates to ≥50 million years ago and that include at least
30 species. Differences between datasets primarily reflect the exclusion of lower-quality assemblies in the
high-stringency subset, which increases the number of universal and complete HOGs while retaining a high
proportion of proteins assigned to orthologous groups.

functions necessary across all species. Ancient HOGs, while also conserved, may include genes that have been differen-
tially retained or lost in some major lineages. In contrast, species- and clade-specific HOGs could represent recent gene
family expansions or lineage-specific adaptations. Where restricted HOGs encode functional proteins, they may contribute
to species-specific traits; however, they also likely arise from methodological artifacts, such as genome annotation errors
or orthology misassignment.

In the full dataset, over half of all the predicted protein-coding genes fell into universal HOGs, reinforcing the idea that
most genes are highly conserved. However, a substantial fraction of the HOGs (~20,000) contained genes from a small
number of species (<30), suggesting either recent evolutionary gains or problems with orthology inference (S4 Fig).
Interestingly, some HOGs classified as ancient were present in only a few species, and it seems probable that many of
these "sparse" HOGs reflect fragmented assemblies or annotation inconsistencies, rather than true biological patterns.
This highlights the importance of caution when interpreting or analyzing low-representation HOGs in comparative studies.
By classifying HOGs based on their evolutionary conservation and phylogenetic distribution, we provide a framework for
future studies of gene conservation and turnover in Drosophilidae. Universal and ancient HOGs likely represent function-
ally essential genes, while restricted HOGs may point to lineage-specific innovations or methodological challenges. This
classification allows us to distinguish between broad evolutionary patterns and potential technical noise, improving the
reliability of our comparative analyses.

To obtain a well-supported set of orthogroups, and to mitigate potential errors from sparse HOGs, we examined in more
detail those HOGs that contained at least one *Drosophila melanogaster* gene. Given that *Drosophila melanogaster* has
one of the most complete and thoroughly verified gene sets among multicellular eukaryotes, its representation in a HOG
provides additional confidence that the group represents a biologically meaningful gene family rather than an artifact.
We identified 12,151 such HOGs, which were broadly shared across the majority of Drosophilidae species (S4 Fig). This
approach provides greater confidence for the analysis of conserved gene families, and allows us to assess how widely
well-characterized genes are distributed across phylogeny. This work substantially expands the most recently-available

gene-orthology set for Drosophilidae from 36 species [32] to 301 species, offering a more comprehensive understanding of gene families across the entire clade, and we hope that this dataset will be valuable for future studies focusing on comparative genomics, evolutionary biology, and functional genomics within Drosophilidae.

## Phylogenetic inference using BUSCO and HOG genes

Comparative analyses generally use an ultrametric phylogenetic tree describing relationships, and thus the expected covariance in traits, among species [65], as this allows the inferences of "phylogenetic heritability". The most comprehensive molecular phylogeny of Drosophilidae to date encompasses 704 species, but this is based on only 17 reference genes and thus has many deep branches that are not resolved with high confidence [80]. More recently, genome-sequencing of 360 species has enabled a BUSCO-gene tree based on one thousand loci—greatly improving confidence in deeper relationships [33]. However, branch lengths were inferred for that tree using 4-fold degenerate sites, which will tend to underestimate deep branch lengths due to substitution saturation. Here we infer the species tree using both HOG and BUSCO approaches, the first dataset comprising 251 single-copy HOGs, and the second comprising 1,824 BUSCO genes. We found that the HOG tree and the BUSCO tree were highly concordant and showed highly supported relationships for all but five species (S3 File). For example, in the HOG tree *Drosophila fuyamai* was positioned as a sister to *Drosophila carrolli*, *Drosophila rhopaloa*, and *Drosophila prolongata*, whereas the BUSCO tree also included *Drosophila kurseongensis* in this clade. Other conflicting relationships can be found in the S3 File. Most internal branches were well supported in both trees, but in some places, the HOG tree exhibited slightly lower local posterior probabilities, particularly for short branches (S4 File). These discrepancies likely reflect increased discordance due to incomplete lineage sorting [34], as resolving such branches requires a larger number of gene trees.

## Factors affecting annotated gene number and CDS Length

To better understand apparent variation in gene number and CDS length across Drosophilidae, we fitted a phylogenetic generalized linear mixed model [65] to assess the impact of "reference" genome status, phylogenetic distance from the reference, genome size, assembly quality (contig N50), read-type (i.e., long-read versus exclusively short-read), and the availability of RNA-seq data on these two traits (S1 File).

Our analysis found no significant differences in gene number or CDS length between our annotations and the established reference annotations, indicating that our annotations are of comparable quality (gene number: $p = 0.8$, 95% HPD CI [−487, 494]; CDS length: $p = 0.67$, 95% HPD CI [−0.05, 0.004]). However, species lost an average of *ca.* 16 genes for each extra million years of divergence from their liftover reference ($p < 0.001$, CI [−23.2, −7.7]), while on average CDS length increased by just one nucleotide per million years divergence from reference ($p = 0.002$; CI [0.2, 1.5]; Fig 2; S1 File). This reflects the increased challenges associated with lift-over between more divergent genomes, but—likely because lift-over was only one source of data among many—the effect is relatively small. Repeating the analyses on the high-stringency subset of 215 genomes produced qualitatively identical results, with only minor changes in effect size (15 fewer genes and one nucleotide difference in mean CDS length; S1 File).

In the full 301 species dataset, "Read type" (i.e., long-read versus exclusively short-read) emerged as the strongest overall predictor of gene content. Assemblies based on short-read data alone contained *ca.* 1,000 more annotated genes than long-read-based assemblies ($p < 0.001$; 95% HPD CI [588, 1,460]), but exhibited shorter CDSs by 60 nucleotides on average ($p = 0.004$; 95% HPD CI [−106.8, −20.2]). Increased assembly contiguity (measured as contig N50) was associated with an average of *ca.* 13 *fewer* predicted genes per mega base-pair increase in contig N50, but this effect was marginal ($p = 0.05$; CI [−26, −0.6]; Fig 2; S1 File). CDS length was unaffected by the assembly contiguity. In the high-stringency subset, where all short-read-only assemblies were excluded, the effect of assembly contiguity on gene number was substantially reduced and no longer statistically supported (*ca.* 8 fewer genes per Mbp increase in contig N50; $p = 0.07$), and CDS length remained unaffected (S1 File). These patterns suggest that short-read assemblies tend to

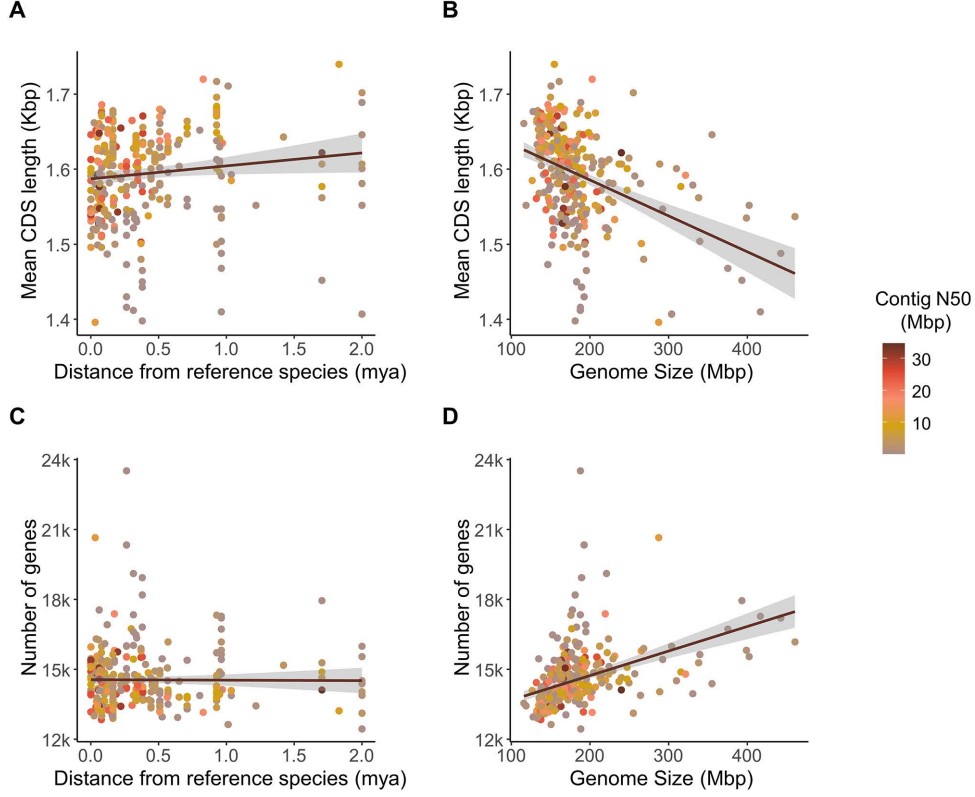

**Fig 2. Variation in gene number and CDS length across Drosophilidae. (A)** Coding sequence (CDS) length increases with phylogenetic distance from the reference species used for lift-over annotation. **(B)** CDS length decreases as genome size increases. **(C)** Gene number remains largely unaffected by phylogenetic distance from the reference. **(D)** Gene number increases with genome size, suggesting a relationship between genome expansion and gene content. In all plots, the points are colored by genome contiguity (contig N50). The numerical data underlying all panels are provided in S4 Table. Fitted lines and 95% confidence windows are derived from a non-phylogenetic linear model and are for illustration only; see main text for phylogenetic mixed model analyses.

inflate gene counts by fragmenting genes into multiple partial models, whereas long-read assemblies and higher contiguity facilitate the recovery of longer, more complete CDSs and reduce spurious gene inflation.

The inclusion of RNA sequencing data unexpectedly *reduced* gene number (by *ca.* 440 genes; $p = 0.002$; CI [−715, −201])—without affecting CDS length ($p = 0.3$, CI [−37.2, 17.1]). This may reflect a reduction in the overall false-positive rate, a reduction in annotated pseudogenes, and the joining of disjunct exons. In the high-stringency subset, the effect of RNA sequencing data on gene number was no longer detectable (S1 File). This suggests that the apparent reduction in gene number associated with RNA-seq in the full dataset primarily reflects improvements in annotation accuracy for lower-quality or more fragmented assemblies, rather than a fundamental dependence on transcriptomic evidence. In high-quality, long-read assemblies, comparative annotation alone appears sufficient to recover most gene models reliably, rendering additional RNA-seq data less influential.

We found that *ca.* 13 genes were on average gained with each additional 1 Mbp of genome assembly size ($p < 0.001$; CI [10.4, 16.5]), but mean CDS length only decreased by less than 1 bp ($p < 0.001$; CI [−0.9, −0.4]) (Fig 2)—that is, larger genome assemblies contained more genes, but these genes were slightly shorter on average. In addition, total CDS length increased only weakly with genome size, by approximately 13 Kbp per Mbp ($p < 0.001$; CI [9, 17]), indicating that increases in gene number are not accompanied by proportional expansion of coding capacity (S1 File). This pattern is

consistent with increased fragmentation of gene models in larger or more repetitive genomes [81], but is also observed in the high-stringency dataset, suggesting that biological processes such as lineage-specific expansion of small gene families may contribute alongside technical effects.

After accounting for these fixed effects, we found little evidence for major differences in gene number or CDS length among *Drosophila* clades. Posterior estimates of differences among internal nodes generally had confidence intervals overlapping zero, suggesting that number of genes and CDS length have remained relatively stable across lineages (S5 Fig). Nevertheless, the *obscura* group had a generally higher inferred gene number and shorter CDS lengths compared to other clades (S5 Fig), which may reflect lineage-specific effects such as variations in karyotype, including transitions among sex chromosomes [82]. Correspondingly, phylogenetic heritability was moderate for gene number (40%; CI [20.5, 63.8]) and low for mean CDS length (9.7%; CI [1.5, 18.7]), indicating that while evolutionary history plays a role, species-specific factors contribute substantially to variation in these traits. Interestingly, when restricting the analysis to the high-confidence subset of 215 genomes, point estimates of phylogenetic heritability increased for gene number (61%; CI [47.4, 77.8]) and slightly decreased for mean CDS length (6.8%; [0.2, 15.3]), although these differences were not significant. This shift suggests that lower-quality assemblies may add noise to gene number estimates, thereby obscuring underlying phylogenetic signal.

Overall, our results indicate that while gene number and CDS length variation occur at the species level, they do not seem to have strong phylogenetic structuring at deeper evolutionary timescales, perhaps predominantly reflecting differences in assembly and annotation rather than long-term evolutionary trends.

## GC composition and codon usage bias in Drosophilidae

To illustrate the potential utility of our new annotations, we analyzed variation in GC content and its relationship with codon usage bias across the full set of 301 drosophilid species [72,83]. Genomic GC content ranged from 21% in *Drosophila neohypocausta* to 49% in *Drosophila nannoptera* (S6 Table), with coding regions, as expected, showing higher GC content (range: 41%–57% GC) than the genome-wide average. GC3 is a widely used proxy for codon bias, and reflects a balance between mutational pressure and selection acting on synonymous mutations [72,83]. In our analysis, GC3 was highly correlated between related species, with an estimated phylogenetic heritability of 1 (i.e., no residual variance that is not captured by the phylogenetic effect), indicating strong conservation within clades and little variation among closely related species. GC3 was also positively correlated with non-coding GC content (Figs 3 and 4; phylogenetic correlation from the PGLMM 0.52; $p < 0.001$; CI [0.41, 0.59]), suggesting that genome-wide mutational biases contribute to both coding and non-coding base composition. Nevertheless, we observed substantial clade-specific deviations; notably the *willistoni* and *saltans* groups, along with subfamily Steganinae, had much lower GC3, whereas the genus *Zaprionus* and the *melanogaster*, *montium*, *obscura*, *ananassae*, *repleta*, and *virilis* species groups exhibited elevated GC3 (S5 Fig). These differences mirror a recent analysis of 29 *Drosophila* species, in which subgenera *Sophophora* and *Drosophila* exhibited distinct codon preferences [72]. Such lineage-specific differences could reflect factors beyond mutation bias or overall GC composition, potentially including selection for translational efficiency or efficacy [84].

To quantify the role of selection in determining codon usage, we estimated the strength of selection on 2-fold codons (quantified by the "*S*" statistic of [76]). Estimates of *S* ranged from 0.24 in *Drosophila pachea* (95% bootstrap interval across genes [0.22, 0.29]) up to 1.08 [0.96, 1.20] in *Drosophila takahashii* (Fig 3). As expected, the *melanogaster*, *montium*, and *ananassae* groups showed elevated GC3 and *S*, confirming stronger selection in favor of GC-ending codons in these groups [72,84,85]. However, the *willistoni* and *saltans* groups—which display low GC3—also showed relatively high *S*, confirming that the AT-bias seen in *Drosophila willistoni* is (at least in part) a result of selection (S5 Fig; [84,86,87]). A similar pattern was also seen in the subfamily Steganinae. Overall GC3 and *S* are negatively correlated across Drosophilidae—indicating that species with higher GC3 are, on average, actually experiencing *weaker* selection on codon usage (Fig 4; phylogenetic correlation from the PGLMM: −0.2; $p < 0.001$; CI [−0.33, −0.10]). Interestingly, a weak *positive*

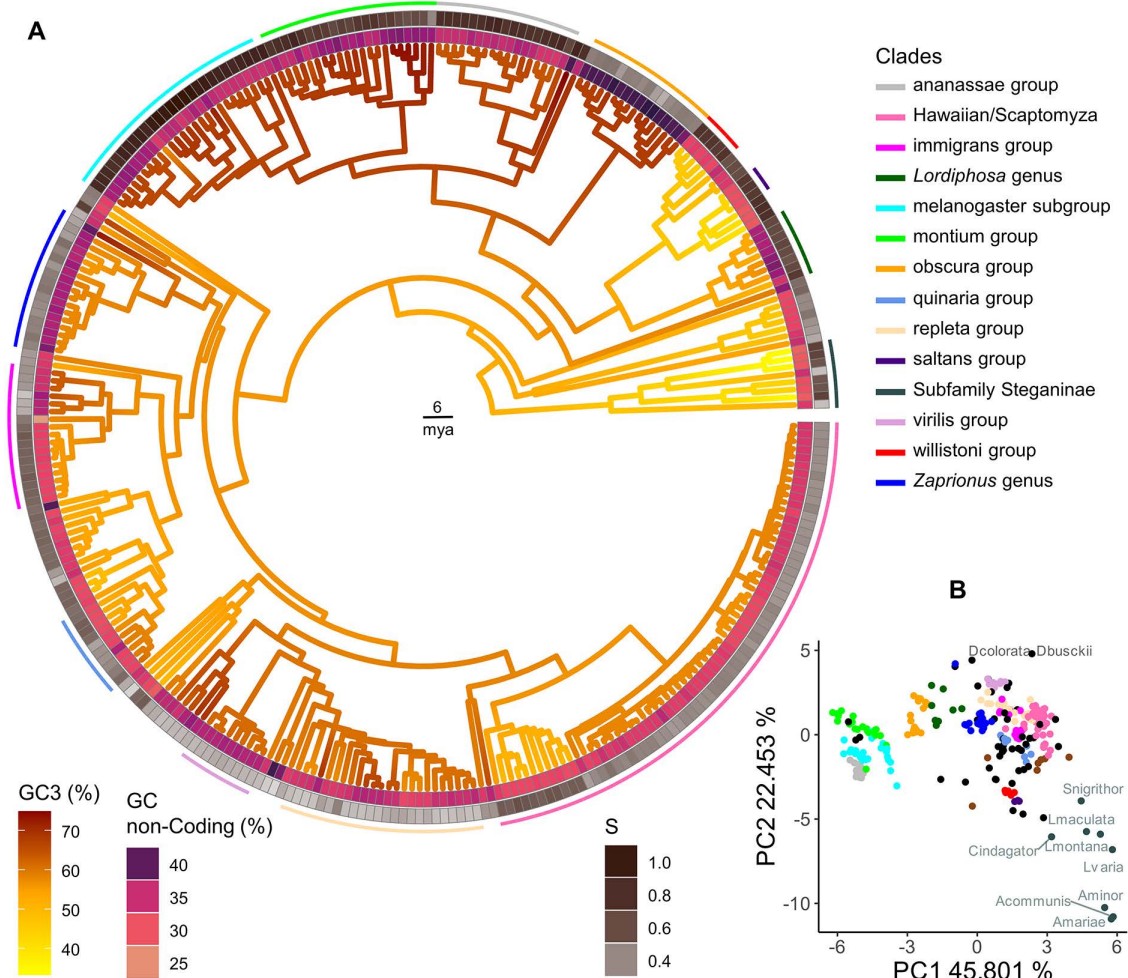

**Fig 3. Codon usage, amino acid composition, and selection on codon usage across Drosophilidae. (A)** Phylogenetic distribution of GC content at third codon positions (GC3), non-coding GC content, and strength of selection on codon usage bias (*S*) across Drosophilidae. The tree is color-coded by clades, and branches are colored according to GC3. Inner ring shows GC content in non-coding regions and outer ring shows strength of selection on codon usage. Major drosophilid species groups are indicated by arcs positioned outside the outermost tile layer. **(B)** Principal component analysis (PCA) of amino acid usage across species, showing that closely related species exhibit similar amino acid composition patterns. The numerical data underlying both panels are provided in S6 Table.

correlation between *S* and genome size (Fig 4; phylogenetic correlation from the PGLMM 0.15; *p* = 0.02; CI: [0.03, 0.32]) suggests that species with larger genomes tend to experience slightly stronger selection on codon usage—in contrast to what might be expected under relaxed constraint in species with small effective population size [88].

## Amino acid composition

To investigate whether the variation in codon usage is associated with variation in amino acid composition, we analyzed the relative proportions of all 20 amino acids across the annotated proteins of Drosophilidae. In general, it is thought that amino acid usage is influenced by a combination of mutational biases, translational selection, and functional constraints—but genome-wide nucleotide composition has been shown to play a significant role in shaping amino acid frequencies [83,89]. Our principal component analysis (PCA) revealed that more closely related species share more similar amino acid usage

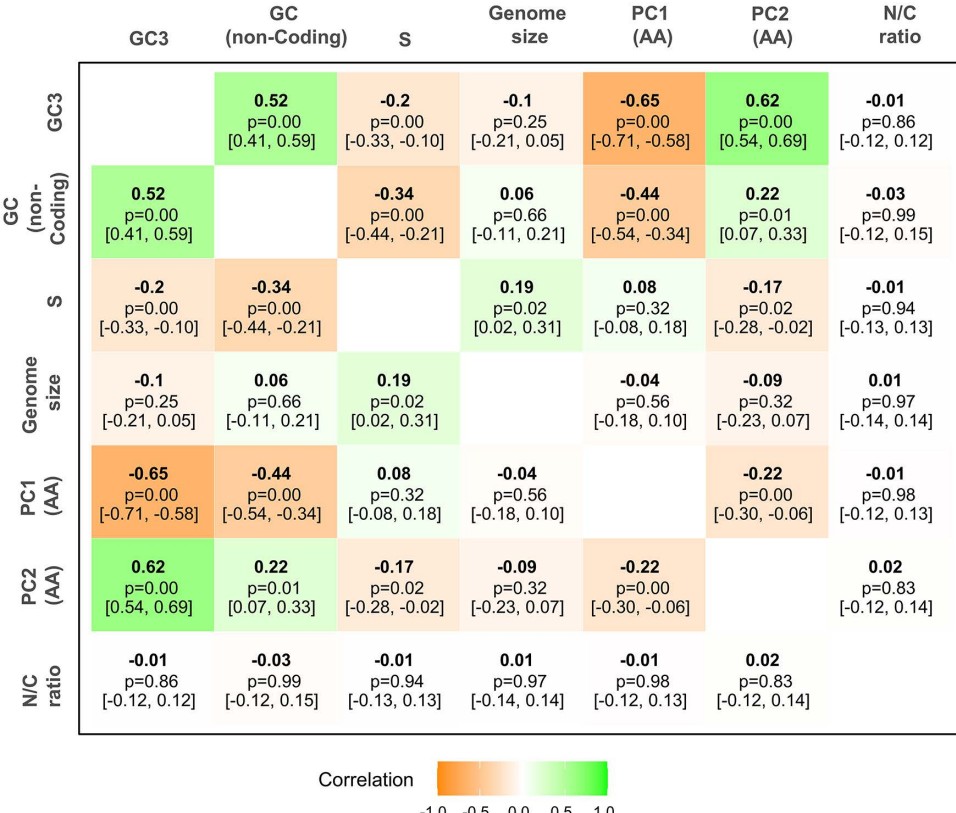

**Fig 4. Correlation matrix of codon usage, genome features, and amino acid composition.** Pairwise phylogenetic correlations between GC content at third codon positions (GC3), GC content in non-coding regions, strength of selection on codon usage bias (S), genome size, principal components PC1 and PC2 of amino acid usage, and nitrogen-to-carbon (N/C) ratio of amino acids. Correlations are derived from the posterior distribution of the phylogenetic variance–covariance matrix and represent evolutionary covariation among traits after accounting for shared ancestry. Values shown correspond to posterior modes; uncertainty is reported as 95% highest posterior density intervals. Strong correlations (positive or negative) are highlighted in green and orange, respectively, indicating relationships between nucleotide composition, codon usage, and amino acid preferences. The numerical data underlying this figure are provided in S7 Table.

patterns (Fig 3). Principal component 1 primarily separated species based on GC content in the codons, with high-GC3 genomes enriched for GC-rich amino acids (Pro, Gly, Ala, Arg) and low-GC3 genomes enriched for AT-rich amino acids (Asn, Tyr, Ile; see Fig 5). To assess whether the patterns were linked to biochemical properties of the amino acids, such as N/C ratio or amino acid essentiality (as measured in *Drosophila melanogaster*; [90,91]), we examined the remaining PCA loadings. However, we found no clear patterns, suggesting that other factors, such as protein structure or functional constraints, may play a small role in shaping variation in amino acid use among species of Drosophilidae (Fig 5).

## Conclusions

This work, to generate standardized, simultaneous multi-species coding DNA sequence annotations across 301 species of Drosophilidae, forms part of an ongoing community effort working toward a comprehensive genomic study of the entire family [33]. We envisage that these new annotations, orthology assignments, and multiple sequence alignments will provide a valuable resource for both single-gene and genome-wide evolutionary studies. And, along with future updates as new genomes are sequenced, this resource will support future research in studies of adaptation and functional genomics within this key model clade.

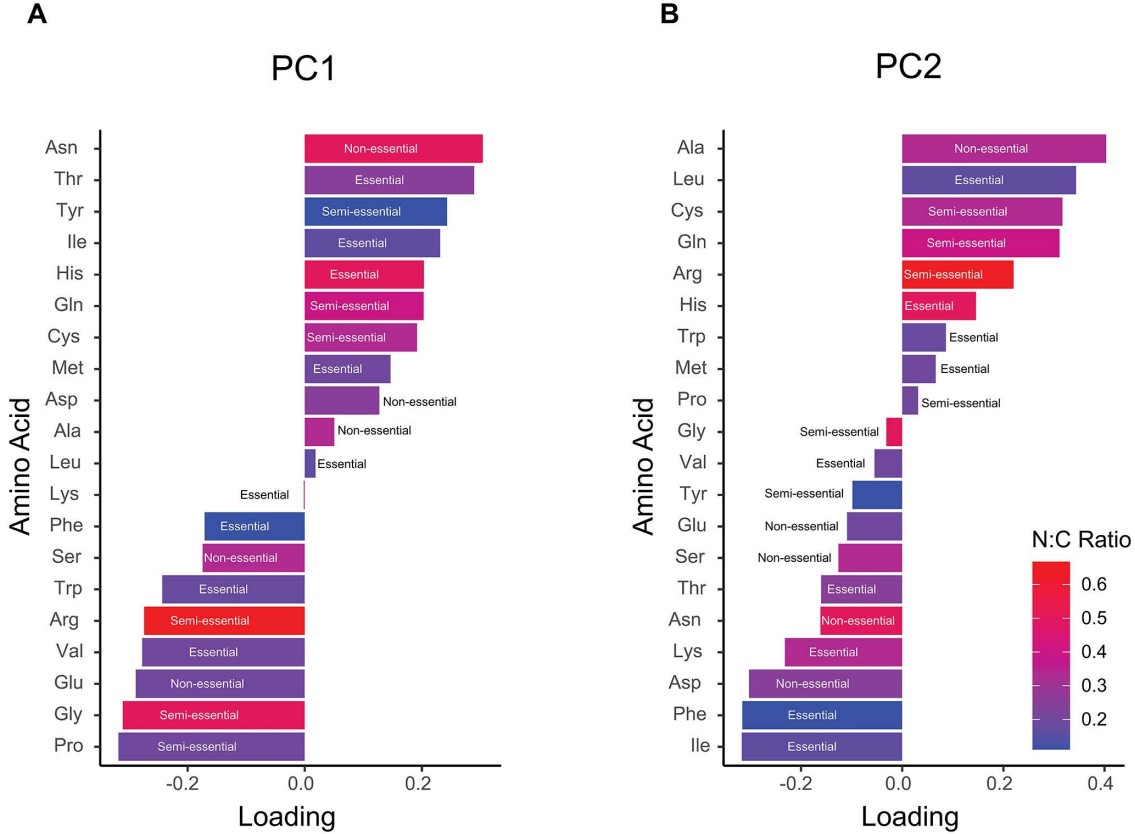

**Fig 5. Principal component analysis (PCA) loadings of amino acid usage.** Bar plots show the loadings of individual amino acids on the first two principal components: **(A)** PC1 and **(B)** PC2. Amino acids are coloured according to their nitrogen to carbon (N:C) ratio, with higher ratios in red and lower ratios in blue. Essentiality categories (essential, semi-essential, non-essential) are indicated alongside each bar. Positive and negative loadings reflect the relative contribution of each amino acid to the corresponding principal component. The numerical data underlying both panels are provided in S8 Table.

## Supporting information

**S1 Fig. Comparison of coding sequence annotations generated by CAT-BRAKER and BRAKER3 alone relative to RefSeq annotations.** Panels show CDS-level precision **(A)**, recall **(B)**, and Jaccard similarity **(C)**, quantified based on pairwise CDS overlap with RefSeq gene models. The numerical data underlying all panels are provided in S9 Table. (TIF)

**S2 Fig. BUSCO completeness scores for genome assemblies and corresponding annotated protein sets across drosophilid species, illustrating concordance between assembly-level and annotation-level recovery of conserved single-copy orthologs. (A)** Annotation-level BUSCO completeness plotted against gene number, showing a weak negative association between completeness and the number of predicted genes. **(B)** Parallel-axis dot plot comparing genome-level and annotated protein-level BUSCO completeness for each species. Species with less than 5% difference are colored gray. **(C)** Relationship between genome-level and annotated protein-level BUSCO completeness, demonstrating a strong positive correlation. Fitted lines and 95% confidence intervals in panels (A) and (C) are derived from non-phylogenetic linear models and are shown for illustrative purposes only. The numerical data underlying all panels are provided in S5 Table. (TIF)

**S3 Fig. Overview of orthology assignments across 301 Drosophilidae species.** The figure shows a time-calibrated phylogenetic tree with ancestral reconstruction of protein-coding gene number mapped onto branches. Tip labels are coloured by the reference species used for comparative annotation within each clade; stars denote reference species, and filled versus open circles indicate the presence or absence of RNA-seq data, respectively. A concentric tile layer indicates the percentage of genes assigned to orthogroups for each species. The outer stacked bar layer shows the number of genes belonging to hierarchical orthologous groups (HOGs) with different levels of species representation (>99%, 75%–99%, 50%–75%, <50%), as well as species-specific HOGs. The numerical data underlying this figure is provided in S4 Table.
(TIF)

**S4 Fig. Classification of Hierarchical Orthologous Groups (HOGs) by phylogenetic depth and species representation. (A)** Number of HOGs plotted against the estimated age of their most recent common ancestor (MRCA, in million years), reflecting the evolutionary depth of gene families. **(B)** Number of HOGs plotted against the number of species in which they are present. **(C)** Number of ancient HOGs (defined as having an MRCA ≥50 million years ago) plotted against species representation. **(D)** Number of HOGs containing at least one *Drosophila melanogaster* gene plotted against the number of species represented. This figure was generated using the HOG summary tables available from the Zenodo repository (https://doi.org/10.5281/zenodo.15016917) and the time-calibrated species phylogeny provided in S5 File.
(TIF)

**S5 Fig. Posterior estimates of genomic and annotation features reconstructed at the most recent common ancestors (MRCAs) of major drosophilid species groups.** Panels show inferred values for gene number **(A)**, mean CDS length **(B)**, GC3 content **(C)**, and strength of selection (S) on codon usage **(D)**, estimated using phylogenetic mixed models. The R model objects used to generate these estimates are provided in S1 Data.
(TIF)

**S1 Table. SRA accession numbers for RNA-seq datasets used in genome annotation.**
(XLSX)

**S2 Table. *Drosophila melanogaster* genes ranked by overall expression level (FPKM).**
(XLSX)

**S3 Table. Expression category assignments for HOGs based on *D. melanogaster* gene expression ranks.**
(XLSX)

**S4 Table. Summary statistics for genome assemblies, genome annotations, and orthology assignments across species.**
(XLSX)

**S5 Table. BUSCO and OMArk completeness assessments for genome annotations.**
(XLSX)

**S6 Table. Estimates of codon usage bias metrics (GC3, GC content in non-coding regions, and selection strength S) across species.**
(XLSX)

**S7 Table. Posterior modes and 95% HPD intervals for phylogenetic correlations among codon usage bias, nucleotide composition, genome size, and amino acid properties.**
(XLSX)

**S8 Table. Principal component loadings for amino acid usage, including nitrogen-to-carbon ratios and essentiality categories.**
(XLSX)

**S9 Table. CDSs Precision, Recall, and Jaccard similarity between CAT-BRAKER and BRAKER only annotations compared to RefSeq annotations.**
(XLSX)

**S1 File. MCMCglmm model summaries and phylogenetic heritability's for full set and high-stringency subset.**
(HTML)

**S2 File. Ultrametric species phylogeny inferred using 251 hierarchical orthologous groups (HOGs).**
(PDF)

**S3 File. Dendrograms comparing species phylogenies inferred from BUSCO and HOG-based datasets.**
(PDF)

**S4 File. HOG and BUSCO phylogenetic trees annotated with posterior probabilities from Bayesian analyses.**
(PDF)

**S5 File. Newick files of species trees generated in this study.**
(ZIP)

**S1 Data. R MCMCglmm model objects generated and used in this study.**
(ZIP)

## Acknowledgments

We wish to thank members of the University of Edinburgh Institute for Ecology and Evolution for the collaborative provision of shared computational resources, James Galbraith for help with EarlGrey and TEstrainer, and Eric Lai and Garima Setia for feedback on missing gene models. We would particularly like to acknowledge the contributions of the broader *Drosophila* research community, whose collaborative efforts in genome sequencing have made this work possible.

## Author contributions

**Conceptualization:** Pankaj Dhakad, Darren J. Obbard.

**Data curation:** Pankaj Dhakad, Bernard Y. Kim.

**Formal analysis:** Pankaj Dhakad, Darren J. Obbard.

**Funding acquisition:** Pankaj Dhakad, Dmitri A. Petrov, Darren J. Obbard.

**Methodology:** Pankaj Dhakad.

**Supervision:** Darren J. Obbard.

**Writing – original draft:** Pankaj Dhakad, Darren J. Obbard.

**Writing – review & editing:** Pankaj Dhakad, Bernard Y. Kim, Dmitri A. Petrov, Darren J. Obbard.

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
