## [Editor Report · Decision Letter 0]

28 Apr 2025

Dear Dr Dhakad,

Thank you for submitting your manuscript entitled "Comparative gene annotation of 304 species of Drosophilidae" for consideration as a Methods and Resources Article by PLOS Biology. Please accept my sincere apologies for the delay in getting back to you as we consulted with an academic editor about your submission.

Your manuscript has now been evaluated by the PLOS Biology editorial staff, as well as by an academic editor with relevant expertise, and I am writing to let you know that we would like to send your submission out for external peer review.

Once your full submission is complete, your paper will undergo a series of checks in preparation for peer review. After your manuscript has passed the checks it will be sent out for review. To provide the metadata for your submission, please Login to Editorial Manager (https://www.editorialmanager.com/pbiology) within two working days, i.e. by Apr 30 2025 11:59PM.

Kind regards,

Richard

Richard Hodge, PhD

rhodge@plos.org

PLOS

---

## [Decision Letter · Decision Letter 1]

25 Jun 2025

Dear Dr Dhakad,

Thank you for your continued patience while your manuscript "Comparative gene annotation of 304 species of Drosophilidae" was peer-reviewed at PLOS Biology as a Methods and Resources article. Please accept my sincere apologies for the delays that you have experienced during the peer review process. Your manuscript has now been evaluated by the PLOS Biology editors, an Academic Editor with relevant expertise, and by two independent reviewers. Please note that we had recruited a third referee to review the manuscript but they are now very late. Therefore, we decided to move ahead with two reviewers to avoid any further loss of time.

In light of the reviews, which you will find at the end of this email, we would like to invite you to revise the work to thoroughly address the reviewers' reports.

As you can see, both reviewers are generally positive about your gene annotation resource and agree that the manuscript provides a potentially sufficient advance as a Resource Article. However, Reviewer #2 raises important concerns about the heterogeneity in genome quality and notes that either low quality genomes are excluded or further validated to improve the robustness and utility of the resource. The reviewer notes that the genome quality threshold is low and suggests that several high-quality assembly/annotation combinations can be used to validate the quality of the annotation procedure. In addition, the reviewer raises concerns about the inclusion of Drosophila subspecies in the assembly.

Given the extent of revision needed, we cannot make a decision about publication until we have seen the revised manuscript and your response to the reviewers' comments. Your revised manuscript is likely to be sent for further evaluation by all or a subset of the reviewers.

**IMPORTANT - SUBMITTING YOUR REVISION**

*Re-submission Checklist*

*Published Peer Review*

*PLOS Data Policy*

*Blot and Gel Data Policy*

Best regards,

Richard

Richard Hodge, PhD

rhodge@plos.org

REVIEWS:

Reviewer #1: Dhakad et al. present the gene annotation of publicly available genome assemblies of 304 species from the family Drosophilidae, using the Comparative Annotation Toolkit (CAT) and BRAKER3, while also including publicly available RNA-seq and protein evidence in their annotation workflow. They supplement the description of their annotation resource by analysing GC and codon usage bias, as well as amino acid composition. Their extensive gene annotation, using sound methodology, in such a large number of genome assemblies is indeed a valuable resource for researchers connected to evolutionary biology and the broader Drosophila research community.

Below are some specific minor comments:

Figure 1: The colours are so similar that the species/groups are hard to tell apart. It doesn't help that orders are completely different.

I don't really see any substantial differences in the violin plots in Fig. 1A, so they mostly clutter the figure without providing much information.

Altogether Fig. 1A is very hard to read. Also, I wasn't able to find the alternative version of the tree in S1 File (…was expecting a figure)

Figure 3B: The overlapping labels are partly unreadable. They should not overlap.

Figure 4: PC1 (AA) at top right should be PC2 (AA), i.e. PC1 label present twice. Also, what type of correlation is it? Pearson's r?

S5 Figure: X-axis labels are shifted to the right and non-aligned (looks messy and does not make interpretation easier)

Line 69: "they typically don't take advantage of closely related species" - why? Isn't it up to user to select protein sequence sets from closely related species or not, using these tools?

Line 207: Why did you prefer the CDS annotation from CAT?

Line 384: A "but" too much?

Just a side note: Looking at the inverse relationship between mean CDS length with genome size and gene number with genome size (Fig 2, S5 Fig.) I wonder if total CDS sequence length would be quite constant across species, regardless of genome size, which might suggest more fragmented annotation in larger genomes? The authors mention annotation artefacts and assembly quality as possible reasons already though.

Reviewer #2: Summary of Manuscript

This manuscript presents a comprehensive comparative gene annotation study of 304 species in the Drosophilidae family. The authors used the Comparative Annotation Toolkit (CAT) and BRAKER3, incorporating RNA-seq and protein evidence, to generate protein-coding gene annotations. They employed a phylogenetic approach to improve consistency and accuracy in annotations and orthology assignments.

Using phylogenetic mixed-model analysis, they found moderate phylogenetic heritability for gene number and CDS length, suggesting that both evolutionary history and species-specific factors influence these traits. To demonstrate the utility of their annotations, they investigated codon usage bias and amino acid composition across Drosophilidae. Their findings indicate that codon usage correlates with overall GC content and evolves slowly, but is also strongly influenced by selection.

The authors pitch this effort as part of an ongoing community effort to improve genomic resources for Drosophilidae, aiming to facilitate research in comparative genomics, including studies on new gene origins, genome size variation, and evolutionary forces shaping gene content and structure.

My impressions

I think it's important to note that this is an expansive, time-consuming project with a very large scope. It is a well-considered and self-evidently time-consuming project. I congratulate and thank the authors for their contributions. It's equally important to note that this work also aspires to be a resource that will be widely used by the community. In light of that, I feel that, on its own terms, this manuscript is suitable in scope and impact for PLoS Biology. What remains for discussion is whether it achieves its goals to an appropriate level. I am very enthusiastic about this manuscript and I am hopeful that it will lead to a resource that will accelerate research for the target community. However, I do have a substantial number of substantive constructive criticisms. In my view, these criticisms must be addressed to ensure the manuscript lives up to its ambitious goals. The current state of the work does not accomplish this. The reason I am being so critical in this area is that I foresee that this manuscript, if published in its current form, will become the de facto annotation resource used for Drosophila until something better comes along, which may be for quite a while. Due to the high integrity of the authors themselves, it is clear that using such a resource will be replete with many caveats on a genome-to-genome basis. In my assessment, this substantially limits its utility. Many users will either shy away from using it in areas where it is clearly a bit rougher, or even worse, may use it with less sophistication and care than is clearly warranted for many genomes and will come to erroneous conclusions as a result.

I think the primary problem is quite straightforward: heterogeneity in genome quality (i.e., assembly quality, gene evidence presence/absence, transcriptome quality, etc). I think the variation in quality is very often cited in the authors' own discussion as a reason for some apparently anomalous observations. To be clear, this variation in quality is a feature of the existing resources the authors are relying on and not in their direct control.

My suggestions are a only a little less simple:

1) exclude more dubious genomes

2) validate the genomes that they maintain as meeting a base level of quality higher than what they currently provide

While this seems fairly vague, I think the authors will know they've achieved it when they aren't focusing so much on explaining anomolous observations in their resource by referring to poor assembly quality.

* I think it's important that QC metrics accompany all genomes that the authors opt to use. The authors have done several, but not all of these:

- heterozygosity/ploidy (GenomeScope)

- Base level quality scores / phred / QV (Merqury)

- Contamination

- Contiguity (Contig N and L stats, probably more than just N50)

- Completeness/Level of Duplication (e.g., all BUSCO stats, particularly don't hide high duplication by only showing the total complete category)

* Manual QC

When the genomes come from sources where the QC isn't understood or isn't very comprehensive, it is a good idea to do these for this resource. For example, in many assemblies, there are idiosyncratic reasons why some important procedures may not be adhered to. Some examples come to mind. In D. miranda, the authors include the Neo-X and the Neo-Y, which is very young and inflates the number of apparent duplicates. In some genomes, some authors may include alternate haplotypes without communicating this to GenBank, and as a result, known althaps are mixed in with the regular genome, also artificially inflating duplicates. To combat this, it would useful to ensure that all genomes without a clearly articulated QC procedure (i.e., Kim et al. 2024 is more clearly articulated than many other genomes) should undergo at least de-duplication and contamination purging to be consistent.

* Y chromosomes sometimes throw monkey wrenches into things, especially if they are new or otherwise have interesting biology. It would be really useful to account for this when known and practicable, though I'm sympathetic that doing this generically is way out of scope. I have in mind neo-Y systems like D. miranda and D. albomicans.

* The authors seem to be following a rubric for discussion of large patterns that goes something like this: whenever something out of the ordinary crops up in (sparse HOGs, e.g.), the idea of annotation error or misassembly is offered as a potential explanation. I'd much rather see a dataset the authors are sufficiently confident in that unusual observations offer legitimately interesting biology because the most likely artifacts have been removed. It's far less interesting for the authors to be constantly reminding readers that the observations they are bringing to our attention may not be biologically meaningful due to heterogeneous assembly/annotation quality. I'm very supportive of the authors bringing such limitations to the reader's attention. But at the same time, if there are that many caveats about this resource, is it really ready to serve as a resource yet? If the caveats loom this large, then using it will be an exercise in caveat emptor rather than a convenient resource.

* Alternate haplotypes, duplicates, and neo-sex chromosomes can all lead to copies of the same basic gene model with genetic variation between them. This is often simply interpreted as duplication, when, in fact, there are three separate processes that lead to this. The authors should incorporate this basic fact about assessing assemblies into account.

* RefSeq usage: There are two issues I see.

1. The usage of RefSeq is hard to understand:

L162 "We selected 37 ‘reference’ species for lift-over annotations based on the completeness and

quality of their genomes, as indicated by RefSeq annotations [36]."

L184 "To perform the genome annotations, we first prepared the reference annotations and extrinsic

‘hints’ for use in CAT [22]. RefSeq annotation files were converted using the “convert_ncbi_gff3”

script provided by CAT, and the resulting GFF3 files were validated with the “validate_gff3”

script to ensure compatibility. We then employed three modes of AUGUSTUS [16] in CAT: two

based on transMap projections (AugustusTM/R) that project annotations from reference

genomes onto target genomes, and one using ab-initio and comparative gene predictions

(AugustusCGP) guided by extrinsic hints [57]."

I don't understand whether RefSeq annotations actually appear in your resource as the annotation

being used or whether they merely feed the annotation software as hints that themselves lead to a

separate annotations. Please clarify.

2. The authors are missing an important feature of using a well-studied genus like Drosophila. There are

several high-quality assembly/annotation combinations that could be used to validate the quality of the

annotation procedure. For example, the authors could compare, for example, Drosophila innubila's high quality

RefSeq annotation to a D. innubila annotation derived from the authors' procedure to evaluate their

approach against a gold-standard in a high quality assembly. This could be repeated for selected high quality

annotations. If the authors are already directly using the RefSeq annotations, then obviously this wouldn't be possible

and maybe an approach that annotates these genomes without recycling the RefSeq data directly could work.

If the authors *ARE* simply lumping in RefSeq annotations with their own bespoke annotations, I would

expect a more detailed discussion of comparing annotations derived from different annotation pipelines.

* L190 "We used Comparative Gene Prediction (CGP) parameters trained on 12 well annotated Drosophila species from the Drosophila 12 Genomes Project, based on exon and intron scoring"

The parameters estimated from that 2016 study are themselves based on the original 12 genomes data, which is frankly terrible. This is not a criticism of the paper, but it is from data published in 2007 and collected before that. If these CGP parameters play an important role in annotation, I suggest justifying or removing them. Notably, non-melanogaster genome assembly quality in 2016 and before is really low. It wasn't until assemblies inspired by studies like Kim et al. 2014 (10.1038/sdata.2014.45) started to change that. And it was definitely after the Konig et al. 2106 studty [57].

* L277 "To assess the factors influencing gene number and CDS length across drosophilid species, we

fitted a multivariate phylogenetic mixed model using MCMCglmm [65]. Our model included gene

number and mean CDS length as response variables, allowing us to analyze their (co-)variation

with respect to predictors such as distance from reference, RNAseq availability, status as a

liftover reference, assembled genome size, and assembly scaffold N50."

It seems like Scaffold N50 is a poor co-factor compared to contig N50. A garbage assembly can be scaffolded into one with a long scaffold N50 with relative ease, and yet the underlying contigs will be a poor representation of the genome. Moreover, scaffolding short contigs is error-prone compared to scaffolding long contigs. And in any event, do all assemblies even have scoffold N50s? Also, it seems like many assemblies are likely from Illumina alone. I think adding something that captures whether or not long reads were used would explain a lot of the variance.

* L123 Minimum N50 of 50kb is way too low in my opinion. I was surprised that a higher threshold wasn't used. I personally consider Drosophila genomes with N50s below 10Mb to be suspect, though I could live with a 1Mb threshold. I would also like to see BUSCOs above 95%.

* L496. No one metric is a reliable proxy for assembly quality, and certainly not N50. Indeed, "assembly quality" is a broad term that encompasses both accuracy of bases and accuaracy of the reconstruction of the structure. It can also include contiguity, of course. I'd like the authors to acknowledge that contiguity isn't the only salient metric of assembly quality.

* L499-520. This is very descriptive and speculative. The authors are generating a lot of hypotheses with their speculations. Why not test some of them?

* Discussions of quality. L381-411

There are at least three discussion points here that seem ripe for reinterpreation. They probably also argue for minor reanalysis or exclusion of assemblies in some cases.

1) Confusion surrounding Drosophila pseudoobscura subspecies leads to inclusion of a bad assembly

Short version: The authors included two Drosophila pseudoobscura assemblies when they should have included only the RefSeq assembly.

Longer version: In the manuscript, the authors comment on outliers in terms of CDS length. They highlight problems with one D. pseudoobscura assembly, noting the unusually short CDS length of GCA_000001765.3. However, in the methods, the authors suggest two relevant criteria by which they selected genomes, including RefSeq genomes and genomes from Kim et al. 2024. In both RefSeq and Kim et al. 2024, the only Drosophila pseudoobscura genome is GCF_009870125.1. In Table S4, the authors report two different D. pseudoobscura genomes (corresponding to the two accessions above). This is extremely confusing. As it happens, both are samples from Mesa Verde and are therefore not only both North American, but both are from the same locale. Obviously, neither is D. pseudoobscura bogotana, since that would have to be from Bogota. So, I don't understand what distinction the authors think they are making by calling one D. pseudoobscura and D. pseudoobcura ssp. pseudoobscura. Moreover, the nomenclature is internally contradictory and inconsistent in the supplemental table. Compare columns A-D in Table S4:

Drosophila_pseudoobscura_pseudoobscura 46245 Dpseudoobsc Drosophila_pseudoobscura_pseudoobscura.GCA_000001765.3.rm.fna

Drosophila_pseudoobscura 7237 Dpseudoobscpseudoobsc Drosophila_pseudoobscura.GCF_009870125.1.rm.fna

to the text:

"The mean CDS length for most species (296 of the 304 species) ranged between 1.42 and 1.74

Kbp. The outliers included Drosophila pseudoobscura ssp. pseudoobscura (GenBank

accession: GCA_000001765.3), which exhibits unusually short CDSs (mean length of 1.15 Kbp)

but a total gene count of 14,546 but close to the family median of 14,249 genes (Fig 1 and S4

Table). This observation raises the possibility that many gene models in such assemblies may

be fragmented, incomplete, or represent short repetitive elements, possibly reflecting issues

with assembly quality or annotation [79, 80]."

I'm familiar with the major high quality genome assemblies in Drosophila. Suffice it to say there is a reason why RefSeq did not select GCA_000001765.3 as the representative genome for the species. As a consequence, please drop the GCA_000001765.3 accession in favor of the RefSeq assembly (GCF_009870125).

2) Inclusion of two Drosophila americana subspecies assemblies (GCA_030788265.1 and GCA_019972375.1). The subspecies distinction may actually be relevant here, but then again, maybe not (GCA_030788265.1 doesn't report the locale, so it's hard to even guess which subspecies it is). In any event, it seems to me that using a shaky subspecies distinction derived from Genbank records to split hairs between two assemblies isn't well justified. Here, the course of action is less clear than above as one assembly appears to be more contiguous (N50) while the other appears to be more complete (BUSCOs). I thought it relevant to bring this to the attention of the authors, though. I think the inclusion of subspecies is far less useful than species and so I encourage the authors to pick one assembly and go with it rather than include both under the current subspecies ambiguity. Personally speaking, for this type of project, I wouldn't even include two equally high quality assemblies per species, even if they happen to be from clearly distinct subspecies. Incomplete lineage sorting between good species is hard enough without throwing subspecies into the mix.

3) The authors state:

"For Drosophila miranda it has previously been shown that there are occurrences of gene gain on its neo-Y chromosome"

Drosophila miranda's neo-Y has indeed acquired about 3,000 genes since its formation. Importantly, however, the Neo-Y resulted from a fusion between Muller element C (called Chromosome 3 in D. pseudoobscura) and the ancestral Y, forming a neo-Y and a neo-X. As it happens, Muller C has around 3,000 genes as well. Since this fusion happened so recently, the genes between the neo-Y and neo-X will appear to be duplicates through the lens of naive annotation approaches which rely on the assumption that chromosomes aren't homologs in the recent past. Since both are descendants of an autosomal Muller C, this assumption is violated. As it happens, if you subtract 3,000 duplicated genes from Mahajan and the 3,000 genes from a diverging pair of Muller C neo-sex chromosomes from the number of D. miranda genes, you get about 14,653 genes in D. miranda (assuming 20,653 genes from Table S4). This is extremely close to the mean of the genus (14,543, again according to Table S4). So, D. miranda isn't all that anamalous after all once its biology is understood and accounted for. Might this also be true for other genomes in the dataset?

To confirm my intuition about whether the Genbank assembly analyzed by the authors actually contains the Neo-Y, I spent about 9 or 10 minutes doing a simple bioinformatics experiment:

1. download the D. miranda genome from GenBank

2. filter using bioawk to separate the NW contigs from the NC scaffolds ($name ~ /NC/ or $name ~ /NW/) and redirect them to different fasta files

3. upload them to D-Genies and inspect the dot plot

The striking result is that NC_046676.1 scaffold (representing the Neo-X) aligns to contigs NW_022881603.1 and NW_022881614.1, which correspond to the Neo-Y1 and Neo-Y2 scaffolds in Mahajan et al. 2018, respectively.

In conclusion, gene gain is almost certainly not the sole explanation for the duplications and increased number of annotated genes. Please confirm whether my quick and dirty inference that many of these duplicates are Neo-Y and Neo-X homologs. If so, then they aren't duplicates. They are simply diverging neo-sex chromosomes. Please correct the manuscript accordingly and see if this type of insight can be baked into the resource's curation.

A quick and dirty way of validating D. miranda's assembly in this context is to simply repeat the analysis after witholding the Neo-Y scaffolds, which should be almost exclusively classified as duplicates through naive homology approaches.

In Mahajan's original paper, the relevant numbers are:

"In total, we identified 6,448 genes on the neo-Y, and 3,253 genes on the neo-X, compared to 3,087 genes on the ancestral autosome that gave rise to the neo-sex chromosome."

* Figure 1: The violin plots are too numerous and individually take up too little of the chart to provide readily accessible information. It's a clever idea that I think fails in the actual execution.

* "we first gathered available RNA-seq data to provide transcript evidence that can help resolve ambiguities in gene."

Singular gene?

---

## [Editor Report · Decision Letter 2]

20 Jan 2026

Dear Dr Dhakad,

Thank you for your patience while we considered your revised manuscript "Comparative gene annotation of 301 species of Drosophilidae" for publication as a Methods and Resources Article at PLOS Biology. This revised version of your manuscript has been evaluated by the PLOS Biology editors, the Academic Editor.

Based on our Academic Editor's assessment of your revision, I am pleased to say that we are likely to accept this manuscript for publication, provided you satisfactorily address the following data and other policy-related requests that I have provided below (A-E):

(A) We would like to suggest a very minor edit to the title, as follows. Please ensure you change both the manuscript file and the online submission system, as they need to match for final acceptance:

“Comparative gene annotation and orthology assignments across 301 species of Drosophilidae”

(B) You may be aware of the PLOS Data Policy, which requires that all data be made available without restriction: http://journals.plos.org/plosbiology/s/data-availability. For more information, please also see this editorial: http://dx.doi.org/10.1371/journal.pbio.1001797

-Supplementary files (e.g., excel). Please ensure that all data files are uploaded as 'Supporting Information' and are invariably referred to (in the manuscript, figure legends, and the Description field when uploading your files) using the following format verbatim: S1 Data, S2 Data, etc. Multiple panels of a single or even several figures can be included as multiple sheets in one excel file that is saved using exactly the following convention: S1_Data.xlsx (using an underscore).

-Deposition in a publicly available repository. Please also provide the accession code or a reviewer link so that we may view your data before publication.

Figure 1B, 2A-D, 3B, 4, 5A-B, S1A-C, S2A-C, S4A-D, S5A-D

(C) Please also ensure that each of the relevant figure legends in your manuscript include information on *WHERE THE UNDERLYING DATA CAN BE FOUND*, and ensure your supplemental data file/s has a legend.

(D) Please ensure that your Data Statement in the submission system accurately describes where your data can be found and is in final format, as it will be published as written there.

(E) Per journal policy, if you have generated any custom code during the course of this investigation, please make it available without restrictions. Please ensure that the code is sufficiently well documented and reusable, and that your Data Statement in the Editorial Manager submission system accurately describes where your code can be found. More information on our Code Policy, what and how to share can be found here: https://journals.plos.org/plosbiology/s/code-availability

We expect to receive your revised manuscript within two weeks.

*Published Peer Review History*

*Press*

Best regards,

Richard

Richard Hodge, PhD

rhodge@plos.org

PLOS

---

## [Editor Report · Decision Letter 3]

4 Feb 2026

Dear Dr Dhakad,

On behalf of my colleagues and the Academic Editor, Chris Jiggins, I am pleased to say that we can accept your manuscript for publication, provided you address any remaining formatting and reporting issues. These will be detailed in an email you should receive within 2-3 business days from our colleagues in the journal operations team; no action is required from you until then. Please note that we will not be able to formally accept your manuscript and schedule it for publication until you have completed any requested changes.

PRESS

Best wishes,

Richard

Richard Hodge, PhD

rhodge@plos.org

PLOS
